# Comparative Proteomic Analysis of Proteins in Breast Milk during Different Lactation Periods

**DOI:** 10.3390/nu14173648

**Published:** 2022-09-03

**Authors:** Yifan Zhang, Xiaoxu Zhang, Lijuan Mi, Chuangang Li, Yiran Zhang, Ran Bi, Jinzhu Pang, Yixuan Li

**Affiliations:** 1Key Laboratory of Functional Dairy, Co-Constructed by Ministry of Education and Beijing Municipality, College of Food Science & Nutritional Engineering, China Agricultural University, Beijing 100083, China; 2Key Laboratory of Precision Nutrition and Food Quality, Department of Nutrition and Health, China Agri-Cultural University, Beijing 100083, China; 3Inner Mongolia Mengniu Dairy (Group) Co., Ltd., Beijing 101107, China

**Keywords:** breast milk, bioactive protein, proteomic analysis, lactation periods

## Abstract

Breast milk is an unparalleled food for infants, as it can meet almost all of their nutritional needs. Breast milk in the first month is an important source of acquired immunity. However, breast milk protein may vary with the stage of lactation. Therefore, the aim of this study was to use a data-independent acquisition approach to determine the differences in the proteins of breast milk during different lactation periods. The study samples were colostrum (3–6 days), transitional milk (7–14 days), and mature milk (15–29 days). The results identified a total of 2085 different proteins, and colostrum contained the most characteristic proteins. Protein expression was affected by the lactation stage. The proteins expressed in breast milk changed greatly between day 3 and day 14 and gradually stabilized after 14 days. The expression levels of lactoferrin, immunoglobulin, and clusterin were the highest in colostrum. CTP synthase 1, C-type lectin domain family 19 member A, secretoglobin family 3A member 2, trefoil factor 3 (TFF3), and tenascin were also the highest in colostrum. This study provides further insights into the protein composition of breast milk and the necessary support for the design and production of infant formula.

## 1. Introduction

Breast milk is the most ideal and natural food for infants, which is rich in a variety of important nutrients for the growth and development of infants [1]. Both clinical and epidemiological studies have shown the short- and long-term health benefits of breastfeeding [2]. It has been observed that breast milk contains a variety of bioactive components that affect the gastrointestinal tract [3] and immune system [4] as well as brain development [5]. Moreover, it has been shown that breast milk reduced the incidence of metabolic diseases among infants and prevented obesity and type 2 diabetes [6]. Consequently, the World Health Organization recommends breastfeeding for 6 months, with complementary feeding for a year or more, depending on the wishes of the mother and baby [7]. According to the lactation stage, breast milk can be divided into colostrum (0–6 d), transitional milk (7–14 d), and mature milk (15 d or more). Colostrum, the first breast milk produced within a few days after delivery, is richer in immunoglobulins and cytokines than mature breast milk [8]. Therefore, ingesting colostrum improves the immune function of infants and provides other health benefits. In addition, it has been reported that colostrum secreted by mothers of a younger gestational age is rich in growth factors [9]. The active factors and related mechanisms in breast milk had been studied, but were still incomplete, especially the identification of active nutrients in different lactation periods.

Protein is an important component of breast milk. It not only provides nutrition for infants but also plays a key role in the formation of the infant immune system and the construction of the intestinal flora [10]. One study showed that lactoferrin in breast milk killed bacteria through an iron-independent mechanism by directly interacting with the surface of bacterial cells [11], contributing to the development of a healthy microbiome. In addition, breast milk is rich in immunoglobulins, which provide local barrier protection. Immunoglobulin A (IgA) is the most abundant immunoglobulin and is usually in the form of secreted immunoglobulin A (sIgA) in breast milk [12]. However, both IgA and immunoglobulin M levels of colostrum decreased significantly in both transitional and mature milk [13]. Osteopontin was found to play an important role in infant immune system maturation, intestinal development, and cognitive development. Milk fat globule membrane proteins were shown to have several health-promoting effects, such as anticarcinogenic, antibacterial, anti-inflammatory, and anti-cholesterol activities [14]. These observations indicated that breast milk was rich in bioactive proteins, and there were still many bioactive proteins to be studied. The rapid development of proteomic technology has made it possible to explore more proteins in breast milk, especially the low abundance proteins with important functions and the changes in proteins during the lactation period, which still need to be further studied [15].

It was shown that the protein content in breast milk was affected by the lactation period and changed greatly in the early stage and gradually stabilized in the later stage [16]. Using proteomics technology to study the changes in these proteins has become a research hot spot. Proteomics also helps to better elucidate the functional changes in the different lactation periods of breast milk. A recent study used a data-independent acquisition (DIA) proteomics approach to investigate changes in proteins in breast milk during 6 months of lactation [17]. This study provided strong evidence for breast milk nutritional differences in different lactation periods. However, with the gradual refinement of infant formula and the importance of breastfeeding in the early life of infants [2], the proteome study of early changes in breast milk (0–1 month) is particularly important. The aim of this study was to identify the composition and changes in the breast milk protein in the first month using DIA quantitative proteomics technology, to explore the potential physiological functions of breast milk in colostrum (3–6 d), transitional milk (7–14 d), and mature milk (15–29 d). In addition, the low-abundance proteins in breast milk were characterized to draw a map of breast milk changes during early lactation and to contribute to the accurate development of infant formula. We also quantified the concentrations of five proteins and performed functional validation on RAW264.7 cells with two biologically active proteins to validate the accuracy of our proteomic results.

## 2. Materials and Methods

### 2.1. Sample Collection

Breast milk samples were collected from 18 healthy lactating mothers at 3–6 days, 7–14 days, and 15–29 days by the Jiangsu Women and Children Health Hospital (Appendix A). All samples were collected each time from the same breast using a portable automatic breast pump in the early morning (9 a.m. to 11 a.m.). The breast milk was collected before feeding. Before collecting breast milk, the hands of the collectors should be washed and disinfected with 75% alcohol. The nipples and areolas also need to be disinfected with 75% alcohol. The breast milk of every three mothers was mixed in the same volume as one sample. Breast milk sample collection has been approved by the Capital Institute of Pediatrics. The collection was carried out in accordance with the Declaration of Helsinki. All participants gave informed consent, and all experiments were conducted according to Chinese laws and institutional guidelines.

### 2.2. Protein Extraction and Digestion

Each of the mixed samples was taken 400 μL and added with 1600 μL of 10%Trichloroacetic acid (TCA) and placed at 4 °C for 12 h. The samples were then centrifuged at 4 °C and 14,000× *g* for 20 min, and the supernatant was discarded. The precipitate was washed by adding 300 μL acetone and vibrating for 30 min at 950 rpm. The supernatant of the samples was discarded after centrifugation at 14,000× *g* at 4 °C for 30 min. Subsequently, the resultant pellet was air dried, which was subsequently added with 200 μL of buffer (7 M urea, 2 M thiourea, and 20 mM Tris-HCl, pH 8.0) to redissolve the pellet. Finally, the solution was centrifuged at 25,000× *g* for 15 min at 4 °C, and the supernatant was taken for use. The protein concentration of the supernatant was determined using a Bradford assay kit (Bio-Rad, Hercules, CA, USA).

Protein digestion was performed using the filter-aided sample preparation method with modifications [18]. Briefly, 100 μg protein was dissolved with 50 mM NH_4_HCO_3_, reduced with dithiothreitol at 56 °C for 45 min, and alkylated with iodoacetamide at room temperature for 30 min in the dark. The solution was transferred into a 10k Da ultrafiltration tube (Vivacon 500, Satrorius, Shanghai, China), and spun at 14,000× *g* for 20 min. In total, 50 mM NH_4_HCO_3_ solution was used to wash the protein three times. Two μg trypsin in 50 μL 50 mM NH_4_HCO_3_ was added and incubated at 37 °C overnight. The ultrafiltration tube was spun at 14,000× *g* for 20 min with a new collection tube to collect digested peptides. NH_4_HCO_3_ solution was added into the ultrafiltration tube to wash the digested peptide into the collection tube. The collected solution was diluted with 0.1% formic acid for nano-LC-MS analysis.

### 2.3. Nano-Liquid Chromatographic Analysis

Nano-Liquid Chromatographic separation was achieved with a Waters (Milford, MA, USA) nanoAcquitynanoHPLC. The C18 trap column was Thermo Acclaim PepMap 100 (75 μm × 2 mm× 3 m). The analytical column was homemade with 100 μm I.D. fused silica capillary (Polymicro) filled with 20 cm of C18 stationary phase (Phenomenex, Aqua 3 μm C18, 125 Å). Mobile phase A was 0.1% formic acid in water; B was 0.1% formic acid in acetonitrile. A gradient elution program was used, with mobile phase increases linearly from 1–35% B in 65 min at a flow rate of 500 nL/min.

### 2.4. DIA Analysis by Nanospray Electrospray Ionization Mass Spectrometry

DIA sample analysis was performed on a Thermo Orbitrap Fusion Lumos high-resolution mass spectrometer (Thermo Scientific, Waltham, MA, USA). The mass spectrometry (MS) parameters were set as follows: MS scan range 350–1250 *m*/*z*; RF Lens 60%; MS resolution 120,000, maximal injection time (MIT) 100 ms; automatic gain control (AGC) target 4 × 10^5^. High-energy collision dissociation (HCD) scans mode for MS/MS, parameters were set as follows: HCD collision energy 32%; scan range 200–1250 *m*/*z* at a resolution of 30,000 with MIT 90 ms; AGC target 10^6^.

### 2.5. Protein Identification and Quantitative Analysis

The raw data from the mass spectrometer were preprocessed with Mascot Distiller 2.7 (Matrix Science, London, UK) for peak picking. The database was downloaded as uniprot_homo_sapiens_irt023.fasta. Trypsin was used for cutting and allows up to two missed cuts. Carbamidomethyl (C) was used for fixed modification while oxidized (M), and Phosphorylation (S, T, Y) was used for variable modification. The maximum missed cleavages was two. The MS mass tolerance was 10 ppm; the MSMS mass tolerance was 0.02 Da. Scaffold PTM was used to evaluate the phosphorylation sites of the Mascot search results using the Score algorithm.

### 2.6. Amino Acids Analysis

The amino acid content in the breast milk was determined by a high-performance liquid chromatography-diode array detector. The method was operated according to GB 5009.124-2016.

### 2.7. Target Proteins Identification and Quantification

According to Bobe et al. [19], after protein extraction, the target proteins: α-lactalbumin, immunoglobulin A, κ-casein, αs1-casein, β-casein, and lactoferrin in breast milk were identified and quantified by high-performance liquid chromatography.

### 2.8. Bioinformatics Analysis

Omicshare online software was used for gene ontology (GO) annotation to analyze the annotation function of milk protein. Pathway analysis of the identified milk proteins was performed based on the online Omicshare software using the Kyoto Encyclopedia of Genes and Genomes (KEGG) pathway database. MetaboAnalyst 5.0 software was used for hierarchical clustering on the identified milk proteins. Finally, the protein–protein interaction network was analyzed using STRING software.

### 2.9. Cell Culture and Viability

RAW264.7 cells (murine macrophage) (ATCC, USA) were grown at 37 °C in a 5% CO_2_ incubation. The RAW264.7 cells (5 × 10^3^ cells/well) were cultured in a 96-well plate to evaluate cell viability. The cells were exposed to the presence and absence of clusterin or TFF3 for 12 h. After reacting with the clusterin or TFF3, the cell viability assay was performed by Cell-Counting-Kit-8.

### 2.10. Detection of TNF-α

The RAW264.7 cells were stimulated with LPS (1 µg/mL) for 18 h and then treated with clusterin or TFF3 (5 µg/mL) for 18 h. We measured the TNF-α levels in supernatants using ELISA kits (MEIMIAN, MM-0180R1, Jiangsu, China) as per the manufacturer’s instructions. 

### 2.11. Western Blotting Analysis

The Toll-like Receptor 4 (TLR4, ab13556) and β-actin (ab22048) antibodies were from Abcam. The RAW264.7 cells were washed using 1 × PBS and lysed by lysis buffer (added as phosphatase inhibitor cocktail 2, phosphatase inhibitor cocktail 3). According to the manufacturer’s instructions, the extraction of cytoplasmic and nuclear protein with the kit (Beyond time, Shanghai, China). The protein content was determined using the Bradford assay. Protein extracts were separated by sodium dodecyl sulfate polyacrylamide gel electrophoresis and transferred to polyvinylidene difluoride membranes (Immune-Blot PVDF membrane, Bio-Rad). 

## 3. Results

### 3.1. Comparison of Total Protein and Amino Acid Content of Breast Milk from Different Lactation Periods

Comparing the differences in the protein content of breast milk from different lactation periods, as shown in Figure 1A, the highest total protein content was in colostrum (1.95 g × 100 g^−1^), followed by transition milk (1.44 g × 100 g^−1^), and mature milk had the lowest total protein content (1.35 g × 100 g^−1^), but there were no significant differences between transition milk and mature milk (*p* > 0.05).

The comparison of amino acid content at different lactation periods is presented in Figure 1B. With prolonged lactation, the content of the total amino acids and each amino acid in breast milk decreased. The amino acid levels in colostrum were significantly higher than in transitional milk and mature milk (*p* < 0.05). The amino acid levels in transitional milk were higher than in mature milk, but there was no significant difference in MET or CYS (*p* > 0.05).

### 3.2. Identification and Quantification of the Proteome in Breast Milk

To compare the amount of breast milk proteins shared by or unique to the different lactation periods, the results determined by DIA revealed 2085 different proteins in the milk from the three different lactation periods, of which 2005 proteins in colostrum, 1952 proteins in transitional milk, and 1855 proteins in mature milk were identified and quantified. Of all the proteins identified, 1782 were found in samples from all three periods, and 81, 25, and 24 proteins were unique to colostrum, transitional milk, and mature milk, respectively (Figure 2A). Colostrum contained the highest number of characteristic proteins.

As shown in Figure 2C, the results of principal component analysis (PCA) showed that colostrum at 3–6 days could be separated from transitional milk and mature milk, while transitional milk and mature milk were not separated, indicating that the proteins expressed in colostrum were significantly different from transitional and mature milk. Likewise, as shown in Figure 2B, the hierarchical clustering analysis of proteins in human milk also revealed that there were two main clusters, with colostrum forming a separate subcluster, while transitional milk was not significantly different from mature milk.

### 3.3. Proteins Differentially Expressed in Different Lactation Periods

To screen the differential proteins of breast milk during different lactation periods, the quantitative data were presented in volcano plots, as shown in Figure 3. Volcano plots with −log10 (*p*-value) against log2 (fold change) were used to compare differentially expressed proteins in breast milk during different lactation periods. It was observed that there were 352 differentially expressed proteins in colostrum compared with mature milk, of which 186 proteins were upregulated and 114 proteins were downregulated. There were 352 differentially expressed proteins in colostrum compared with transitional milk, of which 222 proteins were upregulated and 130 proteins were downregulated. There were fewer proteins in mature breast milk compared with transitional milk, a total of 55 differentially expressed proteins, of which 18 proteins were upregulated and 37 proteins were downregulated.

### 3.4. GO Analysis of the Differentially Expressed Proteins in Breast Milk

To define and describe the function of the differential proteins, the proteins differentially expressed in breast milk during the three periods were analyzed by GO annotations and classified by biological processes, cellular composition, and molecular function. As shown in Figure 4, the most common biological processes were cellular process, biological regulation, the response to stimulus, metabolic process, and the regulation of the biological process; the most common cellular components were cell, cell part, extracellular region, and organelle part; and the most common molecular functions were binding, catalytic activity, molecular function regulator, and structural molecule activity.

### 3.5. KEGG Pathway Analysis of the Differentially Expressed Proteins in Breast Milk

To systematically analyze the protein function, as shown in Figure 5, the KEGG pathway analyses of the differentially expressed proteins compared among colostrum, transitional, and mature milk were divided into metabolism, genetic information processing, environmental information processing, cellular processes, organismal systems, and human disease. The main classes were folding, sorting and degradation, signal transduction, transport and catabolism, immune system, endocrine system, and infectious diseases. As for the differentially expressed proteins in colostrum compared with mature milk, the main participating pathways were metabolic pathways, the PI3K-Akt signaling pathway, and the NF-kappa B signaling pathway, whereas in colostrum, compared with transitional milk, the main pathways were the PI3K-Akt signaling pathway, Epstein-Bar virus infection, and systemic lupus erythematosus, as shown in Figure 5D–F. As for in mature milk compared with transitional milk, the main pathways of the differentially expressed proteins were Epstein-Bar virus infection, human cytomegalovirus infection, microRNAs in cancer, and staphylococcus aureus infection.

### 3.6. Protein–Protein Interaction Network Analysis of Differentially Expressed Proteins in Breast Milk

To compare the interactions of differential proteins, the protein–protein interaction network was analyzed using STRING (Figure 6). The results showed that 37 of the differentially expressed proteins in colostrum and mature milk were directly related. Among these proteins, integrin alpha-V was the most interacting protein with eight interacting proteins, followed by fibronectin with five interacting proteins. Among the differentially expressed proteins of colostrum and mature milk, there were 65 proteins that were directly related. In fact, apolipoprotein and serum albumin, both interacting with six proteins, were the strongest interacting proteins, followed by fibronectin, which interacted with five proteins. There was a total of 23 directly related proteins in mature milk and transition milk. Among them, the DNA replication licensing factor, MCM2, and the DNA replication licensing factor, MCM5, had the strongest interactions with five interacting proteins, followed by paxillin with four interacting proteins.

### 3.7. Validation of Proteomic Results

Through the proteome results, we found that the contents of lactoferrin and IgA were high in colostrum and decreased with lactation, and the contents of α-lactalbumin, αs1-casein, and κ-casein did not change. We analyzed these proteins as the validation of the proteomic results (Figure 7). The analysis showed that the concentrations of lactoferrin and IgA in colostrum were significantly higher than in transitional milk and mature milk. The three high-abundance proteins, α-lactalbumin, αs1-casein, and κ-casein, were not significantly different among the groups. There was no difference in the proteomic results.

Moreover, the KEGG results show that the classification of clusterin belongs to the immune function and TFF3 was high in colostrum. We treated LPS-induced RAW264.7 cells with clusterin and TFF3 to verify the inhibitory effect of these two proteins on inflammation. Prior to evaluating clusterin and TFF3 on LPS-induced inflammation in RAW264.7 cells, we performed a cytotoxic assay to select the proper concentration of clusterin and TFF3 for further investigation. As shown in Figure 8A,B, we observed that clusterin and TFF3 were not toxic to cells and promoted cell viability at a concentration of 5 μg/mL. Therefore, we selected a concentration of 5 μg/mL for further experiments. As shown in Figure 8C, the concentration of TNF-α in the cell culture supernatants was increased under the induction of LPS, and the addition of clusterin inhibited the production of TNF-α compared with the LPS-induced group. The expression of TLR4 was increased in the LPS-induced group and was significantly reduced with the addition of clusterin or TFF3 (*p* < 0.001). These results indicate that some bioactive proteins in breast milk have anti-inflammatory effects.

## 4. Discussion

Breast milk is an incomparable food for infants after birth. Its function not only meets almost all the nutritional needs of infants but also promotes the cognitive and behavioral development of infants to the greatest extent [20]. Studies have found that the composition of breast milk is not fixed and varies significantly with the stage of lactation [21]. A recent study examined proteome changes in breast milk at one month, two months, and six months after birth [17]. However, the period of six months included three lactation stages: colostrum, transitional milk, and mature milk, of which colostrum is extremely important for infants. Therefore, this study used a DIA proteomic method to study the changes in the breast milk proteome within one month. As depicted in the Venn diagram (Figure 2A), we found that colostrum contained the most characteristic proteins. Furthermore, the PCA results (Figure 2C) showed that colostrum could be separated from transitional milk and mature milk, and the hierarchical clustering analysis of proteins in human milk also showed that colostrum was a separate subcluster, revealing that the proteins in breast milk changed greatly between 3 and 14 days. At the same time, the proteins in breast milk gradually stabilized after 14 days. This study used DIA proteomics techniques and provided the first data on protein changes in breast milk between 3 and 29 days.

Previous studies have identified that the major whey proteins in breast milk are α-lactalbumin, lactoferrin, osteopontin, immunoglobulin, lysozyme C, and clusterin [22], and the major caseins are β-casein and κ-casein. These, as well as milk fat globule membrane (MFGM) proteins, were included as high-abundance proteins in this study [23]. Among the major whey proteins, the immune-related proteins are lactoferrin, immunoglobulin, and lysozyme C. Through the results of proteomics and subsequent verification results, we found that lactoferrin was significantly higher in colostrum than in transitional milk and mature milk. This can be explained by infants possibly needing more anti-inflammatory and antibacterial-active proteins in the early stage of life. Iron deprivation is the most basic antibacterial mechanism of lactoferrin. Unsaturated lactoferrin protein has a strong iron-binding property, and one study has shown that it can compete with pathogenic microorganisms to bind iron ions, causing pathogenic microorganisms to stop growing or even die due to the loss of the basic element iron required for growth [24]. As the infant’s immune system matures, the level of lactoferrin decreases. Lactoferrin is of great significance to neonates, and studies have shown that lactoferrin plays a preventive role in the onset of neonatal necrotizing enterocolitis [25]. It has been shown that lactoferrin in colostrum (days 1–5) was significantly higher than in transition milk (days 6–14) and in mature milk (days 14–28) [26], which was consistent with the results of our study. A study also determined the lactoferrin content in the breast milk of 10 full-term mothers and found that the lactoferrin content in colostrum, transitional milk, and mature milk was 6.7 mg/mL, 3.7 mg/mL, and 2.6 mg/mL, respectively [27].

In addition to the significantly higher content of lactoferrin in colostrum, our study found that the content of immunoglobulin and clusterin was also higher in colostrum than in transitional and mature milk. sIgA plays an important role in protecting infants against intestinal and respiratory pathogenic microbial infections [28]. In recent years, studies by other scholars have shown that sIgA in human milk can respond to more than 20 environmental antigens dominated by microorganisms, including rotavirus, Escherichia coli, and enteric pathogens such as Vibrio cholerae and Salmonella [29]. Our study found that the content of immunoglobulin heavy constant alpha 1 in colostrum was significantly higher than in transitional and mature milk, which indicated that the sooner the infant receives breast milk, the more conducive breastfeeding is to the immunity enhancement and healthy growth of the infant. As for clusterin, some studies pointed out that it had a regulatory effect on cell proliferation and also modulated the activity of NF-kappa-B transcription [30,31]. As for β-casein, κ-casein, and α-lactalbumin, there was no significant difference in their content at different lactation stages, which was consistent with previous studies [32].

The innate immunity and acquired immunity in infancy are not fully functional and are highly vulnerable to pathogenic bacteria and other harmful factors. The immune system matures with age [33]. The reported immune-related proteins in breast milk are mainly secreted immunoglobulin A, immunoglobulin G, glycoproteins, immunoregulatory factors, and immune cells [34]. According to the KEGG pathway analysis (Figure 5A), 39 different proteins were found to be involved in the immune system process, and most of these proteins were higher in colostrum than in transitional and mature milk. It was worth noting that among these 39 proteins, changes in clusterin levels in different lactation stages were rarely reported. Through our validation of clusterin on RAW264.7 cells, we found that it was immunologically active. Its function was the same classification, as revealed by KEGG. Breast milk intake becomes especially important before an infant’s immune system is mature.

Among the breast milk proteins from the samples of the three considered lactation periods, the expression levels of cytidine triphosphate (CTP) synthase 1, C-type lectin domain family 19 member A, secretoglobin family 3A member 2, TFF3, and tenascin were the highest in colostrum. Our study demonstrated that CTP synthase 1 was expressed at higher levels in colostrum than in the milk of the other two lactation periods. CTP synthase 1 was shown to be involved in de novo synthesis of CTP and had the ability to sustain the proliferation of activated lymphocytes during the immune response [35]. The higher levels of CTP synthase 1 in colostrum indicated its critical role in the development of the infant’s immune system. As for C-type lectins, in innate immunity, C-type lectins not only played a key role in pathogen recognition but also played an important role in the regulation of the immune response. C-type lectins receptors were shown to recognize fungal cell wall β-glucan and mannan, activate downstream signaling pathways, promote immune cells to secrete IFN-γ, IL-6, TNF-α, and other pro-inflammatory cytokines, and initiate adaptive immune responses to clear fungal infection [36]. Secretoglobin family 3A member 2 has been reported to have a role in fetal lung development and maturation [37]. Other studies have found that secretoglobin family 3A member 2 is a small, secreted protein of about 10 kDa that forms dimers and tetramers, has anti-inflammatory, growth factor, anti-fibrotic, and anticancer activities, and can affect various lung diseases [38].

The protein–protein interaction network analysis (Figure 6) showed a large number of interacting proteins. Interestingly, fibronectin appears several times in the protein–protein interaction network. Studies have shown that fibronectin was essential for osteoblast mineralization [39], indicating that colostrum was important for bone development in infants. The GO annotation showed that the most common molecular functions were binding, catalytic activity, molecular function regulator, and structural molecule activity. Meanwhile, the proteins with these functions were highly up-regulated in colostrum and down-regulated in the late lactation stage, reflecting the unique functions of colostrum in these aspects. Studies have shown that TFF3 was involved in the maintenance and repair of the intestinal mucosa and promoted the mobility of epithelial cells during the healing process [40,41]. The results here may also suggest that colostrum played an important role in the health of the infant’s intestinal mucosa. Interestingly, a study showed that TFF3 regulates the immune response by promoting the migration of monocytes, so as to play a role in the repair and protection of mucosa [42], which indicated its efficacy in immunity. This was consistent with our results validated on RAW264.7 cells. It was reported that the protein tenascin was involved in the guidance of migrating neurons and axons during development, synaptic plasticity, and neuronal regeneration, and promoted neurite outgrowth from cortical neurons grown on a monolayer of astrocytes. In addition, tenascin stimulated angiogenesis in tumors. Interestingly, a previous study found that the expression of tenascin in breast milk at one month was higher than that in breast milk at two months and six months [17], while our study found that the expression of tenascin was the highest between 0 and 5 days, which was consistent with the previous study and complemented the previous research. These biologically active proteins were highly expressed in colostrum, indicating that colostrum intake was of great significance to infant growth and development. 

The innovations of this study and other breast milk proteomics studies are that DIA proteomics technology was used in this study, the changes of breast milk proteome within 3–29 days were provided for the first time, and the results were verified at the cellular level.

## 5. Conclusions

In conclusion, the DIA proteomic method can be used to quantitatively study breast milk proteins during different lactation periods. In the investigated breast milk, a total of 2085 proteins were identified and quantified. The results of this bioinformatics study will provide new insights into the physiological functions of these proteins, especially those related to immunity. The study shows the specificity and importance of colostrum. It was revealed that colostrum plays an extremely important role in the early life of infants. In addition, our findings may provide further insight into the protein composition of colostrum, providing necessary support for the design and production of infant formula.

## Figures and Tables

**Figure 1 nutrients-14-03648-f001:**
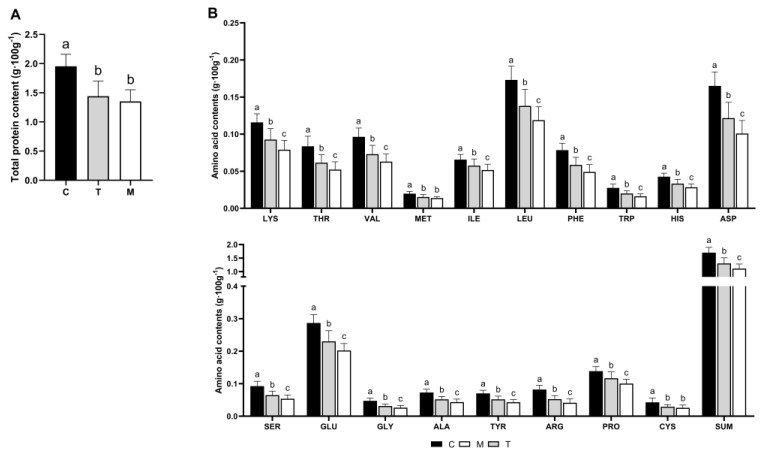
Comparison of total protein contents and amino acid contents in different lactation periods. Different letters represent significant differences in total protein contents in different lactation periods (*p* < 0.05); C, colostrum (3–6 days); T, transitional milk (7–14 days); M, mature milk (15–29 days) (**A**) total protein contents. (**B**) total amino acid contents.

**Figure 2 nutrients-14-03648-f002:**
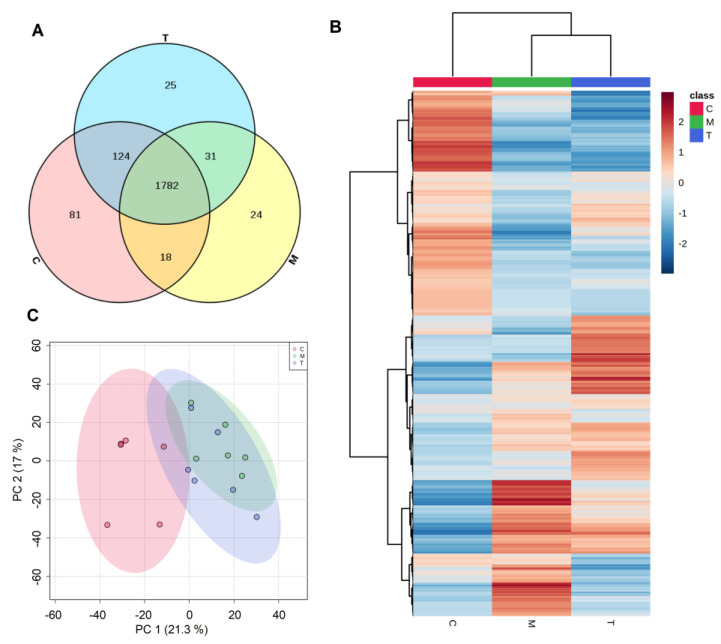
(**A**) Venn diagram analysis (**B**) hierarchical clustering; (**C**) Principal component analysis (PCA) score plot; C, colostrum (3–6 days); T, transitional milk (7–14 days); M, mature milk (15–29 days).

**Figure 3 nutrients-14-03648-f003:**
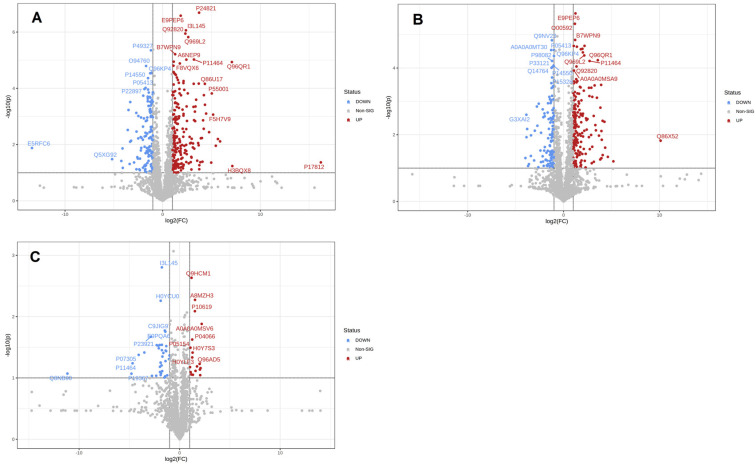
Differential expression of proteins between groups. Volcano plot showing the significance versus fold change in C vs. M (**A**), C vs. T (**B**), and M vs. T (**C**). Down-regulated proteins are shown in red, and up-regulated proteins are shown in blue.

**Figure 4 nutrients-14-03648-f004:**
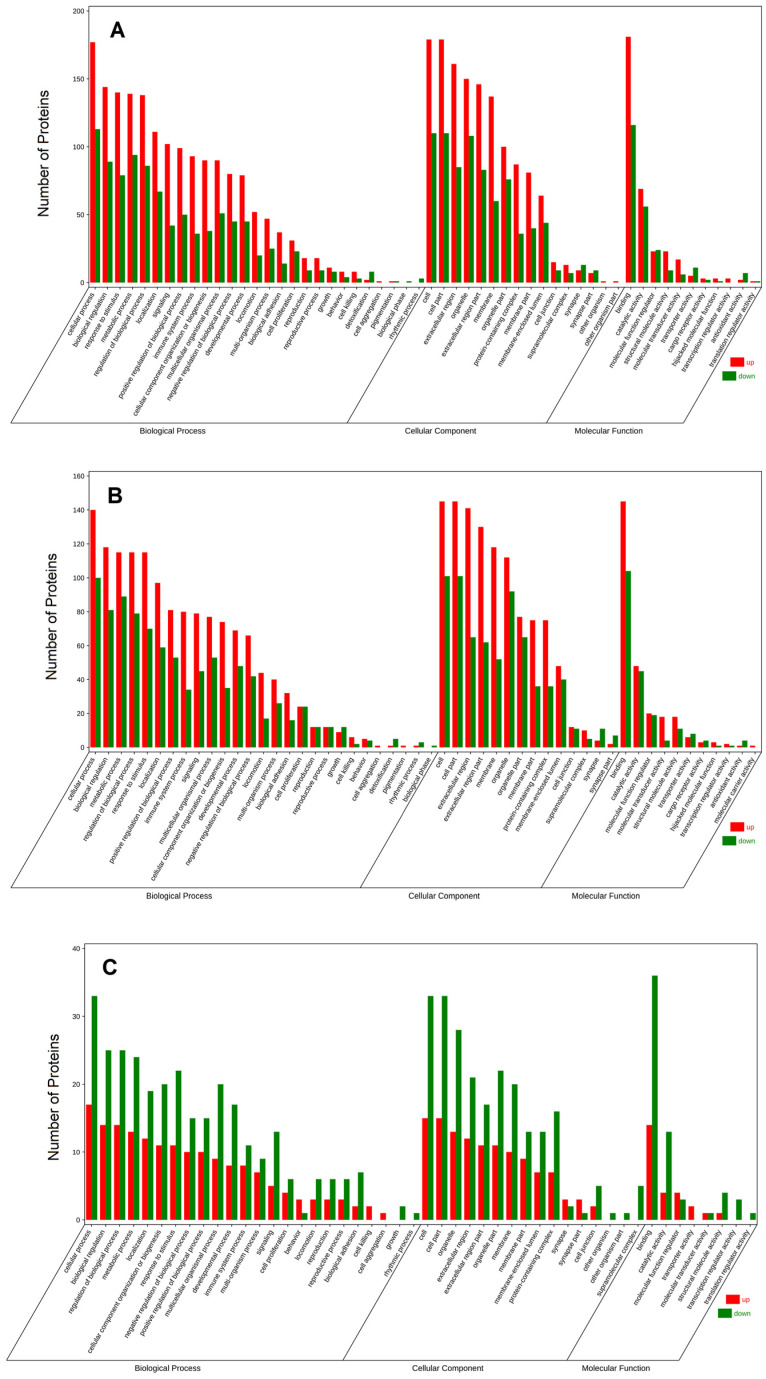
GO annotation of differentially expressed proteins in C vs. M (**A**), C vs. T (**B**), and M vs. T (**C**).

**Figure 5 nutrients-14-03648-f005:**
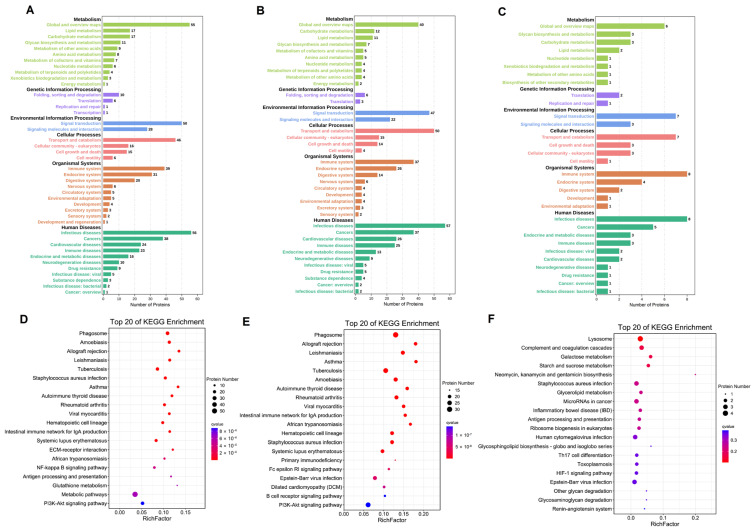
Analysis of differentially expressed proteins in C vs. M (**A**), C vs. T (**B**), and M vs. T (**C**) in KEGG pathway. Top 20 of KEGG enrichment pathway in C vs. M (**D**), C vs. T (**E**), and M vs. T (**F**). The size of the dot represents the number of differential proteins annotated to the pathway. The color of the dot represents the size of the P value.

**Figure 6 nutrients-14-03648-f006:**
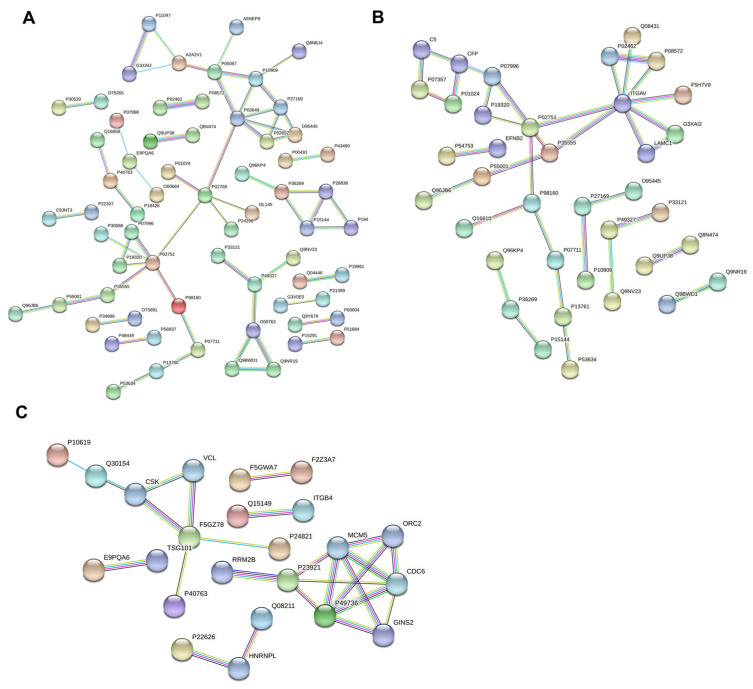
Protein–protein interaction network analysis of differentially expressed proteins in C vs. M (**A**), C vs. T (**B**), and M vs. T (**C**). Each node represents a protein, and each edge represents the interaction between proteins.

**Figure 7 nutrients-14-03648-f007:**
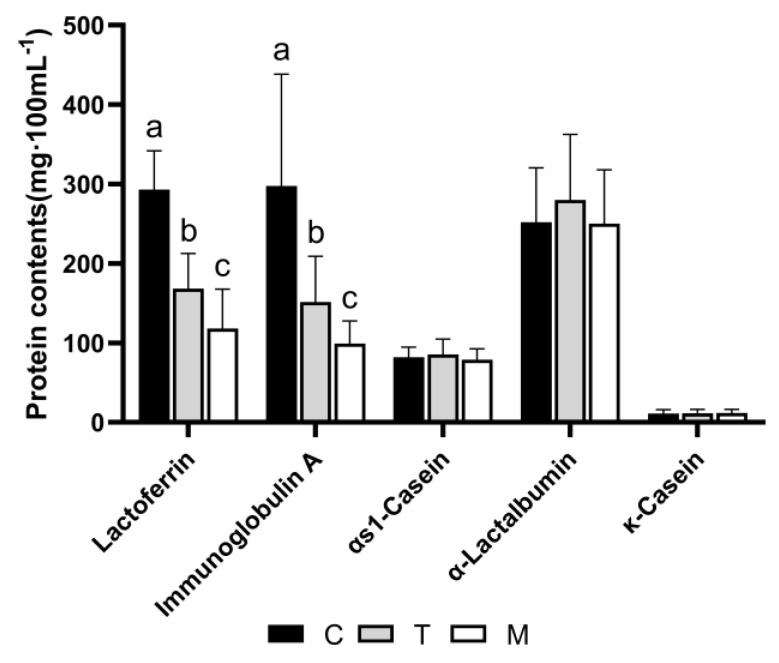
The concentrations of lactoferrin, immunoglobulin A, αs1−casein, α−Lactalbumin, and κ−casein in three different lactation periods. Different letters represent significant differences in protein contents in different lactation periods (*p* < 0.05).

**Figure 8 nutrients-14-03648-f008:**
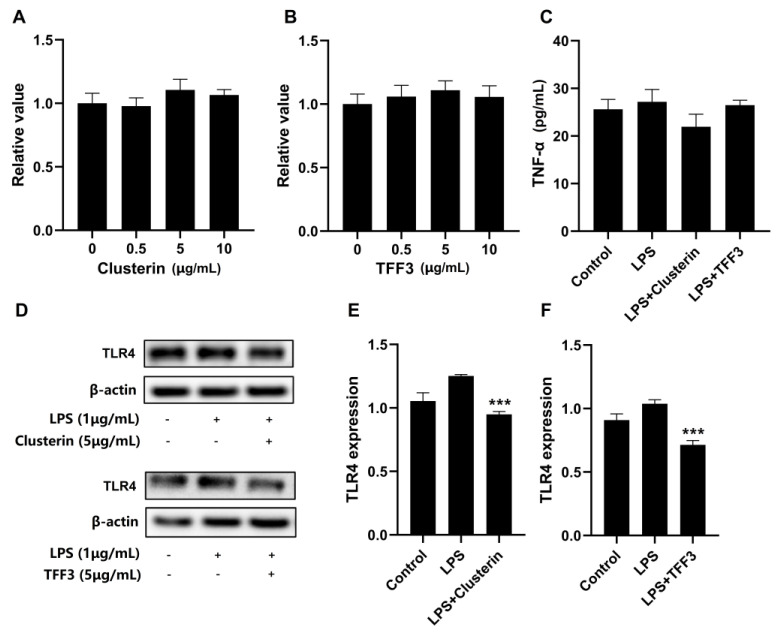
Inhibitory effect of clusterin and TFF3 macrophage inflammation in RAW264.7 cells (**A**,**B**) RAW264.7 cells treated with clusterin or TFF3 (0, 0.5, 5, 10 μg/mL) for 8 h. The cell viability was detected using Cell−Counting−Kit−8 (**C**) RAW264.7 cells were pretreated with clusterin or TFF3 (5 μg/mL) for 8 h after co−incubation with LPS (1 μg/mL) for 8 h. The TNF−α level was detected using ELISA kits. (**D**–**F**) RAW264.7 cells treated with clusterin or TFF3 (5 μg/mL) for 8 h after incubating with LPS (1 μg/mL) for 8 h. The expression of TLR4 was detected using Western blotting. *** *p* < 0.001 versus the LPS induced group.

## Data Availability

Data presented in this study are available on request from the corresponding author.

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
