# Peer review of "Comparative Proteomic Analysis of Proteins in Breast Milk during Different Lactation Periods"

_nutrients, 2022, doi:10.3390/nu14173648_

Round 1
Reviewer 1 Report
Interesting research. Impressive statistical analysis performed. However, samples size was small, it would be good if more samples were collected and analyzed.
It seems that some incorrect citation form in Line 34 to 35.
Sample collection process should be described more detailed. What means "collected hygienically"? (washed hand using soap before milk collection, breasts washed with soap before sample collection process?) Milk collected before/after the baby was fed? Please provide more information.
Did additional information was collected about the participating mothers & babies - maternal age, parity, exclusive or mixed breastfeeding, sex of the baby, birth weight, birth mode etc.? As this co-factors could also impact the results.
Conclusions could be more comprehensive, more emphasizing the differences between colostrum and transitional and mature milk and the significance of colostrum.
Reviewer 2 Report
The manuscript described a study on human breast milk proteome across the first month of lactation. The study was well-designed and executed with advanced techniques. The writing of the manuscript needs improvement; the arrangement of different sections (introduction, methods, results and discussion) should be revised to clarify research gaps, research aims and major outcomes of the study.
Please consider the comments below:
Line 21 (Line 78, 330, 332 and maybe others): colostrum was collected from 3-6 days so it’s not precise to make claims from day 0 or day 1.
Line 34-35: “et al. 2010” should be removed
Line 45-46: “the active factors in breast milk and associated mechanisms are not known…”. I don’t think this sentence is accurate. The authors cited many studies on the bioactive proteins in breast milk and their roles and mechanisms in immune and other functions.
Line 48: This sentence is too complex and becomes incorrect.
Line 54: Please unify the format of IgA in the manuscript. In different parts, “IgA”, “immunoglobulin A”, “sIgA”, and “secreted immunoglobulin A” are used. Once introduced an abbreviation, please use it consistently later in the manuscript.
Further on abbreviations: some abbreviations are not introduced (e.g. FASP L104; DDA, L143), some are introduced but not used (much) later (RP, L114; FA, L119; HPLC, L125, and those in L137-140)
Line 88 and 94: These descriptions seem contradictory. Were every 3 samples pooled or not?
Line 101-102: “The protein concentration of the supernatant”
Section 2.2 and 2.3: It’s confusing why trypsin digestion was described twice (Line 104-115:).
Line 131-132: How were the 10 fractions used for later analysis?
Sections 2.7, 2.8, and 2.9: Please describe in the introduction why these analyses were performed. I was confused until reading the discussion, which is too late. Please also see comments on section 3.7 below.
Line 173: Typo “levelsbin”; please provide manufacturer info for the ELISA kit.
Line 182: What was measured with Western blot should be mentioned in the section title or first sentence, not in the last sentence.
Line 195: What method was used for the amino acid analysis? Also, are the proportions of amino acids different between lactation stages?
Line 204: “amount of proteins” Have the data been normalized based on total protein concentration?
Section 3.7:
1. No method is described for quantifying lactoferrin, IgA, α-lactalbumin, αs1-casein, κ-casein.
2. No proteomic results have been described in the above sections so far on lactoferrin, IgA, α-lactalbumin, αs1-casein, κ-casein, clusterin or TFF3 so it’s very confusing that this section is “Validation of Proteomic Results”. Some results of these proteins are currently in the “Discussion” section below (Line 341, 355-368, etc.), which is a confusing order for readers. One suggestion would be to combine the results and discussion sections, describe differences in the proteome over lactation stages first, and then move to the results on other “validation” analyses. Also, a brief introduction relating to the analyses in Section 3.7 should be added to the introduction section.
Section 4/Discussion:
1. Please add figure references when referring to results presented earlier. E.g. Line 325 “As depicted in the Venn diagram (Fig 2A)…”.
2. Please describe the findings from your study first before discussing other studies. Don’t hide them in a literature review. For example, the paragraph from Line 333 is about lactoferrin. It should start with your results on lactoferrin. (Line 342)“Through the results of proteomics and subsequent verification results, we found that lactoferrin was significantly higher in colostrum than in transitional milk and mature milk.” is the only time that the proteomics result on lactoferrin is mentioned. This is a proteomics study, please highlight your proteomics results first before anything else.
3. There is no discussion of the results in Figures 4 and 6. What do they mean?
4. What is novel about this study compared with other proteomics studies on breast milk? Better highlight in the discussion.
Line 352 “content of colostrum” should be “lactoferrin content in colostrum”
Line 367-368: "consistent with previous studies" Please add references
Line 394: “infected fungi” would mean the fungi is infected; maybe use “fungal infection”
